# Effects of Transcranial Magnetic Stimulation Therapy on Evoked and Induced Gamma Oscillations in Children with Autism Spectrum Disorder

**DOI:** 10.3390/brainsci10070423

**Published:** 2020-07-03

**Authors:** Manuel F. Casanova, Mohamed Shaban, Mohammed Ghazal, Ayman S. El-Baz, Emily L. Casanova, Ioan Opris, Estate M. Sokhadze

**Affiliations:** 1Department of Biomedical Sciences, University of South Carolina School of Medicine-Greenville, 701 Grove Rd., Greenville, SC 29605, USA; Manuel.Casanova@prismahealth.org (M.F.C.); Emily.Casanova@prismahealth.org (E.L.C.); 2Department of Psychiatry & Behavioral Sciences, University of Louisville, 401 E Chestnut Str., #600, Louisville, KY 40202, USA; 3Department of Electrical and Computer Engineering, University of South Alabama, Mobile, AL 36688, USA; mshaban@southalabama.edu; 4BioImaging Research Lab, Electrical and Computer Engineering Abu Dhabi University, Abu Dhabi 59911, UAE; mohammed.ghazal@adu.ac.ae; 5Department of Bioengineering, University of Louisville, Louisville, KY 40202, USA; ayman.elbaz@louisville.edu; 6School of Medicine, University of Miami, Miami, FL 33136, USA; ixo82@miami.edu

**Keywords:** Autism spectrum disorder, evoked and induced gamma oscillations, EEG, TMS, oddball task, reaction time, aberrant and repetitive behaviors

## Abstract

Autism spectrum disorder (ASD) is a behaviorally diagnosed neurodevelopmental condition of unknown pathology. Research suggests that abnormalities of elecltroencephalogram (EEG) gamma oscillations may provide a biomarker of the condition. In this study, envelope analysis of demodulated waveforms for evoked and induced gamma oscillations in response to Kanizsa figures in an oddball task were analyzed and compared in 19 ASD and 19 age/gender-matched neurotypical children. The ASD group was treated with low frequency transcranial magnetic stimulation (TMS), (1.0 Hz, 90% motor threshold, 18 weekly sessions) targeting the dorsolateral prefrontal cortex. In ASD subjects, as compared to neurotypicals, significant differences in evoked and induced gamma oscillations were evident in higher magnitude of gamma oscillations pre-TMS, especially in response to non-target cues. Recordings post-TMS treatment in ASD revealed a significant reduction of gamma responses to task-irrelevant stimuli. Participants committed fewer errors post-TMS. Behavioral questionnaires showed a decrease in irritability, hyperactivity, and repetitive behavior scores. The use of a novel metric for gamma oscillations. i.e., envelope analysis using wavelet transformation allowed for characterization of the impedance of the originating neuronal circuit. The results suggest that gamma oscillations may provide a biomarker reflective of the excitatory/inhibitory balance of the cortex and a putative outcome measure for interventions in autism.

## 1. Introduction

Rhythmic patterns of neural activity, manifested in the electroencephalogram (EEG) as voltage oscillations, have been linked to varied cognitive functions such as perception, attention, memory, and consciousness. The reciprocal interaction between excitation (pyramidal cells) and inhibition (interneurons) during cortical activation provides the genesis for brainwave oscillations [1]. Those brainwaves with the highest frequency, between 30 and 90 Hz, comprise the gamma bandwidth [1,2]. Fast-spiking interneurons that provide for the perisomatic inhibition of pyramidal cells, control the rhythm (clockwork) of these high frequency oscillations [1]. Immunocytochemical characterization of these cells reveals that they express the calcium-binding albumin protein parvalbumin (PV). The high metabolic activity of PV cells, which comprise the largest subgroup of cortical interneurons, makes them highly susceptible to oxidative injury. This pathoclisis helps explain their putative relationship to abnormalities of gamma aminobutyric acidergic (GABAergic) neurotransmission in many psychiatric disorders [3]. 

Reduced numbers of PV-expressing cells have been reported in human postmortem brain samples [4] and animal models of autism spectrum disorder (ASD) (e.g., *Fmr1*, VPA, *Nlgn3,* R451C, and *Cntnap2*) [5]. More significantly, the reduced levels of PV expression correlate with ASD-like behavioral deficits (e.g., sociability, vocalization) and, curiously enough, with symptoms usually ascribed to ASD comorbidities (e.g., pain sensitivity, seizures) [6,7]. Long-lasting reversal of PV (GABAergic) deficits by pharmacologic or cell type-specific gene rescue, normalizes, or at least diminishes, cognitive dysfunction, and social deficits in these animal models [5,8,9]. It is therefore unsurprising that ASD researchers have proposed using gamma-band-based metrics, both a putative “electrophysiological endophenotype” [6] of PV pathology and a metric indicative of the cortical balance between inhibition and excitation [2], as an outcome measure for interventions aimed at targeting the underlying pathology of ASD [10,11,12]. 

Gamma band activity is thought to reflect the mechanism for the integration of information in neural networks within and between brain regions (for reviews see [12,13]). Gamma rhythm is normally defined as EEG band in the frequency range between 30 to 90 Hz (or even higher), although there is an opinion [14] that different frequency sub-bands (e.g., 30–35 Hz, 40–48 Hz, etc.) may have distinct functional significance. Our study focuses on gamma sub-band within 35–45 Hz (so-called 40 Hz-centered gamma [15,16]). Oscillatory activity in the 40 Hz-centered gamma range has been related to Gestalt perception and to cognitive functions such as attention, learning, and memory [17,18]. Binding of widely distributed cell assemblies by synchronization of gamma frequency activity is thought to underlie cohesive stimulus representation in the brain [19,20]. It has been proposed that “weak central coherence” in autism could result from a reduction in the integration of specialized local networks in the brain caused by a deficit in temporal binding that depends on gamma synchronization [21,22,23]. It is important to emphasize that there are distinct functional differences between spontaneous gamma, evoked gamma power and coherence, and event-related induced gamma power and coherence [14]. Sensory evoked gamma coherences reflect the property of modality-specific networks activated by a sensory stimulation. Event-related (or cognitive) induced gamma and its coherences manifest coherent activity of sensory and cognitive networks triggered by and governed by requirements of a cognitive task. In autism, synchronization between these neural networks is abnormal and reflects an imbalance of the excitation/inhibition bias of the cerebral cortex (vide supra, [2]). Studies have shown that resting gamma power appears to be inversely correlated to ASD severity as measured by the Social Responsiveness Scale (SRS) [24]. 

Illusory contour or illusory figure (e.g., Kanizsa figure [25]) perception is a very useful model to study the integration of local image features into a coherent percept, and tests based on several illusory figures were productively used to investigate the impairment of such integration in children with ASD. Brown et al. [22] tested adolescents with autism in an experiment that presented Kanizsa shapes with visual illusions and reported excessive evoked gamma at 80 and 120 ms post-stimulus, in addition to enhanced induced gamma (200–400 ms). Inability to reduce gamma activity would lead to the inability to decide which event requires attention when there are multiple choices. In autism, uninhibited gamma activity suggests that none of the circuits in the brain can come to dominance because too many are active simultaneously [21,22,23]. Abnormalities of gamma synchrony can result in significant cognitive deficits, such as reduced attentional control, and other dysfunctions present in ASD. In addition, EEG recordings during a Kanizsa figure task have shown an overall increase in gamma oscillatory activity in ASD as compared to neurotypicals [22]. These findings are thought to reflect a reduction in the “signal to noise” level due to diminished inhibitory processing [22]. These observations are of clinical significance as several studies have now reported in ASD that abnormalities in gamma oscillations are normalized by low frequency repetitive transcranial stimulation (TMS). This neuromodulatory therapy also provides improvements in both repetitive behaviors and executive functions [10,26,27,28,29]. 

In a recent study comparing ASD and neurotypical controls, spectral analysis of the outer envelope joining the upper peaks of gamma oscillations allowed researchers to characterize the settling time after peak voltage amplitude [30]. At baseline, with no active treatment instituted, the latency of the ringing decay assessed using frequency analysis was significantly diminished in ASD as compared to control subjects. A short ringing time indicates a system whose efficiency of operation or sensitivity is diminished [31]. The oscillations induced by tasks involving the integration of features, as for example in a reaction time tasks using Kanizsa illusory features. Our group has used oddball task paradigms of target classification and discrimination which required a response to target Kanizsa squares among non-target Kanizsa triangles and other non-Kanizsa distractor figures in order to examine event-related potentials (ERP) and amplitude of gamma-band EEG activity [10,12,29]. We reported differences between neurotypical children and children with ASD diagnosis in reaction time and ERP measures as well as amplitude of gamma responses. Furthermore, we reported normalization of ERP responses and improved behavioral symptoms in children with ASD following repetitive transcranial magnetic stimulation (rTMS) treatment [28]. The current study was focused on more advanced analysis of evoked and induced gamma oscillations using the same illusory figure task. The gamma waveforms elicited by this task exhibit a characteristic dampening after peak amplitude in which the outer envelope of successive peaks traces a decay curve that persists until baseline. 

The study used demodulation of gamma oscillations allowing to examine both the envelope of a signal as well as the periodic waveform that carries the same suggesting that resonance behavior, exemplified in the carrier wave may tie neural populations operating at the same frequency. Analysis of the envelope of gamma oscillations can be used to investigate the impedance of involved circuits and the excitatory inhibitory balance of the cerebral cortex [30]. We propose that the metrics of gamma oscillations, ingrained in both its carrier and its envelope, may provide important information contributing to better understanding of functional significance of EEG gamma waveforms.

Despite all of the evidence, the utility of gamma-band related variables as diagnostic biomarkers is currently unexplored, suggesting an urgent need for using gamma oscillation measures as functional markers of response to interventions such as rTMS or other types of neuromodulation. This sensitivity is what allows a system to respond selectively to a given frequency while eliminating others. Brainwave oscillations are not a finely tuned process; amplitude, frequency and phase all vary across individual gamma cycles [32]. In autism, the low sensitivity makes the synchronization between neuronal networks imprecise. The end result is a distortion in the ability to form cohesive perceptual experiences and a reduction in the brain’s ability to provide for nuanced responses to both environmental and social exigencies.

The findings described in the previous paragraphs led the authors to study and compare, in ASD and a neurotypical control population, the metrics for gamma oscillations that describe its envelope. This study expands on previous findings by analyzing the evoked and induced components of gamma oscillations using wavelet transformation. Given the many reports in the literature which translate gamma oscillations to possible behavioral states we also analyzed for possible correlates to aberrant and/or repetitive behaviors in our ASD population post-TMS treatment.

## 2. Methods and Materials

### 2.1. Subjects

Children and adolescents with ASD diagnosis were recruited through referrals from several pediatric clinics. All patients (N = 19 mean age, 14.4 ± 3.61 years old, 5 females) were diagnosed according to the Diagnostic and Statistical Manual of Mental Disorders (DSM-IVTR) and/or DSM-5 [33,34]. Diagnosis of autism was further ascertained with the Autism Diagnostic Interview-Revised (ADI-R) [35]). A developmental pediatrician evaluated the patients, ascertained them to be in good health, had normal hearing, and were willing to participate in lab testing. Participants were excluded if they had a history of seizures, impairment of vision, genetic disorders, and/or brain abnormalities based on neuroimaging studies. Exclusionary criteria for this group were as follows: (a) current diagnosis of any Axis I psychiatric disorder, such as psychosis, bipolar disorder, and schizophrenia; (b) current psychiatric symptoms requiring medication other than those for attention deficit/hyperactivity disorder (ADHD); (c) severe medical, cognitive or psychiatric impairments that would preclude from cooperation with the study protocol; and (d) inability to read, write, or speak English. The EEG test procedures also required the following exclusionary criteria: (1) impaired, non-correctable vision or hearing; (2) significant neurological disorder (epilepsy, encephalitis) or head injury. Subjects enrolled in the study were high-functioning children or adolescents with a full-scale Intelligence Quotient (IQ) of more than 80 according to evaluations using the Wechsler Intelligence Scale for Children, Fourth Edition (WISC-IV, [36]) or the Wechsler Abbreviated Scale of Intelligence (WASI, [37]). Children with an ASD diagnosis who were on stimulant medication were included in this study only if they were taken off medication on the day of the lab visit for testing. Four participants had ADHD diagnosed before they were diagnosed with either autistic disorder or Asperger Syndrome by DSM-IV [33], while one subject was diagnosed by DSM-5 [34] as ASD comorbid with ADHD.

Typically developing children (i.e., control subjects, CNT group, N = 19, 14.8 ± 3.67 years old, 6 females) were recruited through advertisements in the local media. All control participants were free of neurological or significant medical disorders, had normal hearing and vision, and were free of psychiatric, learning, or developmental disorders based on self- and parental reports. Subjects were screened for a history of psychiatric or neurological diagnosis using the Structured Clinical Interview for DSM-IV Non-Patient Edition (SCID-NP [38]). Participants within the control and ASD groups were matched by age, gender, full scale IQ, and socioeconomic status (based on parental level of education and annual household income) of their family. The age of participants was in 9 to 17 years range, with majority of them being adolescents within 11–15 years range, but since according to the National Institutes of Health (NIH) definition the participants under 18 years old are still are categorized as children they are further referred to for convenience as “children” rather than “children and adolescents”.

The study was conducted in accordance with relevant national regulations and institutional policies and complied with the Helsinki declaration. The protocol of the study including informed consent and assent forms that were reviewed and received approval of University of Louisville (Louisville, KY, USA) Institutional Review Board (IRB) (Ethical approval protocol#006.07). Children and their parents or legal guardians received detailed information about the research study specifics, including its purpose, responsibilities, reimbursement rate, risk vs. benefits evaluation, etc. The participants were reimbursed only for oddball tests ($25 for each procedure), and did not receive any reimbursement for the TMS treatment. Investigators provided consent and assent forms to all families who expressed interest in participation in this treatment research study and answered all questions related to the project. If the child and his family member confirmed their commitment and agreed to be part of it, both child and parent signed and dated the consent and assent forms and received a copy co-signed by the study investigator. 

### 2.2. Experimental Task: Visual Oddball with Illusory Kanizsa Figures

The test used in the study was a three-stimuli oddball task with rare illusory Kanizsa squares (target, 25%), rare Kanizsa triangle (non-target Kanizsa, 25%) and frequent non-Kanizsa stimuli (standards, 50%) [25]. Non-target Kanizsa and non-Kanizsa standard stimuli are further referred to as task-irrelevant stimuli. Visual stimuli were presented for 250 ms with inter-trial interval in the 1100–1300 ms range. Subjects were instructed to press a button on a keypad with their right index finger when a target appeared, and not to respond to any of the task-irrelevant stimuli. This task required the processing of both stimulus features (shape and collinearity) for discrimination of targets. Before the test, all subjects had a brief practice block to get familiar with the specifics of the task, make sure that they understood the test requirements, and that they could recognize the target stimulus correctly. There was a total of 240 trials in the study and 20 trials in the practice block. The test took approximately 20–25 min to complete. Participants with ASD diagnosis had at least one lab visit before the test to ensure habituation to the experimental setting and lab environment as well as conditioning to the EEG sensor net. 

### 2.3. Event-Related Gamma Oscillations Recording

The dense-array (128 channel) electroencephalogram (EEG) was recorded with an Electrical Geodesics Inc. Netstation system (EGI-Philips, Eugene, OR, USA). Experimental control (e.g., stimulus presentation, reaction time) was executed using E-prime software (version 1.1, Psychological Software Tools (PST), Inc., Pittsburg, PA, USA). Visual stimuli were presented on a monitor located in front of the subject, while motor responses were recorded with a 4-button keypad (PST’s Serial Box). EEG was recorded with 512 Hz sampling rate, analog Notch (60 Hz, IIR, 5th order) filter and analog bandpass elliptical filters set at 0.1 to 100 Hz range. Electrodes impedance was kept under 40 Ω as recommended by the EGI Netstation manual. Raw EEG recordings were segmented off-line spanning 200 ms pre-stimulus baseline and 800 ms epoch post-stimulus. EEG data was screened for artifacts and all trials that had eye blinks, gross movements and other artifacts were removed using Netstation artifact rejection tools [39]. Other details of our experimental procedure and EEG data acquisition, pre-processing and analysis can be found in our prior studies using the same methodology [26,27,28,29,40,41]. Stimulus-locked dependent EEG variables for the frontal (F3, F4, F7, F8) and parietal (P3, P4, P7, P8) sites-of-interest referenced to vertex (Cz) and nasion as a ground were used for gamma oscillation analysis. Analysis of gamma oscillations was performed on trial-by-trial basis. Data set was not re-referenced for average reference frame but rather accepted trials were left with initial vertex reference to avoid distortion of gamma waves. Evoked gamma was analyzed with 40–160 ms window and induced gamma within 180-400 ms window post-stimulus following ranges recommended by prior studies using similar designs and comparable stimuli [12,14,18,19,20,22]. 

### 2.4. EEG Analysis

#### Method Description

EEG signal *x (t)* represented by 500 time samples per trial was analyzed using a Continuous Wavelet Transform (CWT) where the Morlet wavelet is used as an analysis wavelet. The CWT for the EEG signal (i.e., *X (*
τ*,s)*) is defined as follows:(1)$$X\left(\tau ,s\right)=\frac{1}{\sqrt{s}}\int_{0}^{\infty }x\left(t\right)\left(\frac{t-\tau }{s}\right)dt$$
where τ is the Morlet wavelet that is a continuous function in both time and frequency, is the time shift of the wavelet, and *s* is scale of the wavelet. In fact, the scale reveals crucial information about the signal. The smaller the scale, the more compressed the wavelet in the time domain, and the more focus on high frequencies (i.e., rapidly changing details). However, the larger the scale, the more stretched the wavelet in the time domain, and the more focus on low frequencies (i.e., slow changing coarse features). X(τ,s) is then calculated at 500 different scales spanning various frequencies constituting the EEG signal.

Further, in order to obtain the gamma wave, a frequency-localized inverse CWT is used where the coefficients X(τ,s) corresponding to the frequency range (35–45 Hz) are extracted and an inverse CWT is applied to the extracted coefficients X(τ,s) as follows:(2)$$G\left(t\right)=\frac{1}{C}\int_{s1}^{s2}\int_{0}^{\infty }\frac{1}{s^{2.5}}X\left(\tau ,s\right)\varphi \left(\frac{t-\tau }{s}\right)d\tau ds$$
where *G (t)* is the continuous time gamma wave, φ is the dual function of such that both functions are orthonormal and *C* is a constant calculated from the mother wavelet. Also, *s*_1_ corresponds to the highest frequency in the gamma wave frequency band while *s*_2_ corresponds to the lowest frequency in the frequency band.

Next, we will define several characteristics of the discrete time representation of the gamma wave (i.e., *G (n)*) such as zero crossings, peaks, major peak, latencies, and areas of the left and right halves of the positive envelope of the gamma wave. 

Zero crossings are defined as the time location where the sign of the gamma wave changes from the positive to the negative and vice versa. Zero crossings can be classified into two types; upward zero crossings and downward zero crossings. An upward zero crossing is found when the gamma wave changes from a negative to a positive value while a downward zero crossing is located when the gamma wave changes from a positive to a negative value. Both upward zero crossing ($Z_{upward}$) and downward zero crossing ($Z_{downward})$ are defined as follows:(3)$$Z_{upward}=n_{0}\;s.t.\;G\left(n_{0}\right)<0\;and\;G\left(n_{0}+1\right)>0$$
(4)$$Z_{downward}=n_{1}\;s.t.\;G\left(n_{1}\right)>0\;and\;G\left(n_{1}+1\right)<0$$

A peak is defined as the maximum value of the set of values within the time interval defined between an upward zero crossing and a downward zero crossings. The amplitude of the peaks ($P\left(\bar{n}\right)$ ) can be represented using the following mathematical equation:(5)$$P\left(\bar{n}\right)=\max\;G\left(n\right)|Z_{upward}<n<Z_{downward}$$

Further, the amplitude of the major peak (*P_M_* ) of the evoked and induced gamma waves can be defined as the maximum value of all the peaks ($P\left(\bar{n}\right)$) located within the corresponding timeframe.
(6)$$P_{M}=\max\;P(\bar{n)}|{\bar{n}}_{1}<\bar{n}<{\bar{n}}_{2}$$
where ${\bar{n}}_{1}$ and ${\bar{n}}_{2}$ are time location of the starting and ending peaks of the evoked or induced gamma waves. 

Slopes are calculated between the major peak and the time location of the lowest peak values mentioned above. A positive slope ($S_{+}$) is defined between the lowest peak at the beginning of the gamma wave time interval (${\bar{n}}_{1}$) and the major peak while a negative slope ($S_{-}$) is defined between the major peak and the lowest peak at the end of the gamma wave time interval (${\bar{n}}_{2}$).
(7)$$S_{+}=\frac{P_{M}-P\left({\bar{n}}_{1}\;\right)\;}{\bar{n}-{\bar{n}}_{1}\;}$$
(8)$$S_{-}=-\frac{P_{M}-P\left({\bar{n}}_{2}\;\right)\;}{\bar{n}-{\bar{n}}_{2}\;}$$

Latencies are defined as the time difference between the location of the major peak and the location of the lowest-amplitude peaks at the beginning and at the end of the gamma wave time intervals (i.e., ${\bar{n}}_{1}\;$ and ${\bar{n}}_{2}\;$ respectively).
(9)$$Latency_{1}=\bar{n}-{\bar{n}}_{1}$$
(10)$$Latency_{2}=-\left(\bar{n}-{\bar{n}}_{2}\right)$$

The areas of the left and right halves for the positive envelope of the gamma wave can be approximated using the following equation:(11)$$A_{L}=\frac{1}{2}\left(Latency_{1}\times P_{M}\right)$$
(12)$$A_{R}=\frac{1}{2}\left(Latency_{2}\times P_{M}\right)$$

### 2.5. Transcranial Magnetic Stimulation

Repetitive TMS was administration using a Magstim Rapid device (Magstim Co., Sheffield, UK) with a 70-mm wing span figure-eight coil. For the identification of resting motor threshold (MT) for each hemisphere the output of the magnetic stimulator was increased by 5% steps until a 50 μV deflection of electromyogram (EMG) or a visible twitch in the First Dorsal Interosseous (FDI) muscle was detected in at least 2 or 3 trials of TMS delivered over the motor cortex controlling the contralateral FDI. Electromyogram was recorded with a psychophysiological monitor C-2 J&J Engineering Inc. with USE-3 software (version 3, J&J Engineering Inc., Poulsbo, WA, USA) and Physiodata applications (Physiodata, Inc., Winslow, WA, USA). 

The rTMS was administered on a weekly basis with the following stimulation parameters: 1.0 Hz frequency, 90% MT, 180 pulses per session with 9 trains of 20 pulses each with 20–30 s intervals between the trains. Initial six weekly rTMS session were administered over the left dorsolateral prefrontal cortex (DLPFC) and the next 6 were over the right DLPFC, while the additional 6 treatments were done bilaterally (evenly over the left and right DLPFC). The procedure for stimulation placed the TMS coil 5 cm anterior, and in a parasagittal plane, to the site of maximal FDI response as judged by the FDI EMG response. To ensure better positioning of the TMS coil a swimming cap was used on a head of subject. The location for TMS stimulation was performed with anatomical landmarks [42,43] that approximate the scalp region used for F3 and F4 EEG electrode placements in the 10-20 International System. Motor threshold was detected for the left hemisphere during session 1, for the right hemisphere at session 7, and for both hemispheres at session 13. 

We selected 90% of the MT based on reports from prior studies where low frequency rTMS was used for the stimulation of DLPFC in whole range of neurological and psychiatric disorders [44,45,46,47]. we decided to have stimulation power below MT as a safety precaution meant to lower the probability of seizures in this study population. The decision to use low frequency (below or equal 1 Hz) magnetic stimulation was based on the finding that at this frequency range rTMS exerts an inhibitory influence on the stimulated cortex [48]. Visual oddball tests in the ASD group were conducted with a week before the 18 session-long rTMS course and within a week after the completion of the course of intervention.

### 2.6. Behavioral and Social Functioning Evaluation

For evaluation of social and behavioral functioning caregiver (parent or guardian) reports were used. Participants in the ASD group were evaluated before TMS course and within a week following treatment. Aberrant Behavior Checklist (ABC, [49]) is a rating scale to assess Irritability, Lethargy/Social Withdrawal, Stereotypy, Hyperactivity, and Inappropriate Speech based on parent/caregiver report. Repetitive Behavior Scale—Revised (RBS-R [50]) is a caregiver completed rating scale to assess stereotyped, self-injurious, compulsive, ritualistic, sameness, and restricted range. 

### 2.7. Statistical Analysis 

Repeated measure ANOVA was the primary model for statistical analyses of subject-averaged evoked and induced gamma oscillation metrics (positive and negative areas of gamma oscillation envelope), motor response, and behavioral questionnaires data. Dependent behavioral variables were RT, omission and commission response rate, and total accuracy. Dependent stimulus-locked evoked and induced gamma variables were positive/ascending and negative/descending areas values at pre-determined frontal (F3, F4, F7, F8) and parietal (P3, P4, P7, P8) EEG sites of interest. The within-participant factors for analysis of TMS effects were the following: *Stimulus* (Target Kanizsa (TRG), Non-target Kanizsa (NTG), non-Kanizsa (NOK)), *Hemisphere* (Left, Right), and *Time* (ASD Baseline, ASD Post-TMS). Comparisons for ASD and CNT groups used also *Stimulus* × *Hemisphere* × *Group* (CNT, ASD pre-(ASD), ASD post-TMS (TMS)) factors. Post-hoc analyses were conducted where appropriate using one-way ANOVA. For behavioral rating scores, a *Treatment* (pre- vs. post-TMS) factor was used. Histograms with normal distribution curves along with skewness and kurtosis data were obtained for each dependent variable to determine normality of distribution and appropriateness of data for ANOVA tests. All dependent variables in the study had normal distribution. Greenhouse-Geisser (GG) corrected p-values were employed where appropriate in all ANOVAs. For the estimation of the effect size, we used a Partial Eta Squared (ƞ_p_^2^). IBM SPSS (version 26.0, Armonk, NY, USA) and Sigma Stat statistical packages (version 9.0, Systat Software, Inc. San Jose, CA, USA) were used for data analysis.

## 3. Results

### 3.1. Behavioral Responses (Reaction Time and Accuracy)

There were no group differences in reaction time (RT) between ASD and neurotypical (CNT) children groups. The ASD group had more commission errors (11.22 ± 15.48 % in ASD vs. 1.55 ± 3.48% in CNT, F_1,36_ = 7.06, *p* = 0.012) and more omission errors (2.12 ± 2.72% vs. 0.55 ± 1.18%; F_1,36_ = 5.31, *p* = 0.027). 

Effects of TMS on RT to targets were not significant (*p* = 0.51, n.s.). There were post-TMS group differences in accuracy, namely in total percentage of errors (13.35 ± 16.84% pre- vs. 3.16 ± 3.09% post-TMS, F_1,36_ = 6.73, *p* = 0.014). Group differences in commission error percentage were also statistically significant (11.22 ± 15.48 vs. 2.23 ± 2.51% post-TMS, F_1,36_ = 6.24, *p* = 0.017). 

### 3.2. Behavior Evaluations Post-TMS

Repetitive Behavior Scale Outcomes: Repetitive behavior subscales (RBS-R, [50]) showed group difference for Stereotype Behavior and T-score. Stereotypy scores decreased from 5.53 ± 3.85 to 3.26 ± 2.57 (F_1,36_ = 4.53, *p* = 0.040); Compulsive Behavior scores decreased from 3.95 ± 2.82 to 2.00 ± 1.82, F1,36 = 6.39, *p* = 0.016), and Total Repetitive Behaviors T-score decreased from 23.74 ± 14.54 to 5.21 ± 9.84 (F_1,36_ = 4.47, *p* = 0.041). 

Aberrant Behavior Checklist Outcomes: Two of the ABC [49] subscales showed significant post-TMS differences. Irritability scores decreased from 11.74 ± 8.66 to 6.63 ± 5.65 (F_1,36_ = 4.62, *p* = 0.038); and Hyperactivity scores from 17.47 ± 15.60 to 8.79 ± 7.34 (F_1,36_ = 4.79, *p* = 0.035). Lethargy/Social Withdrawal scores showed trend to statistically non-significant decrease from 7.89 ± 4.89 to 5.89 ± 4.45, F_1,36_ = 1.60, *p* = 0.241 (n.s).

### 3.3. Evoked and Induced Gamma Oscillations

*Evoked gamma.* Positive slope area (ascending half-envelope area): *Stimulus* had main effect at the frontal sites F7/F8 (F_2,55_ = 17.63, *p* < 0.001, ƞ_p_^2^ = 0.391), *Stimulus* (TRG, NTG, NOK) × *Group* (CNT, ASD, TMS) interaction was significant (F_2,55_ = 4.56, *p* = 0.002, ƞ_p_^2^ = 0.139). Main effects of *Stimulus* and *Stimulus* × *Group* interactions at F3/F4 did not reach a significant level. At the parietal P3/P4, main effect stimulus was significant (F_2,55_ = 13.85, *p* < 0.001, ƞ_p_^2^ = 0.201). *Stimulus* × *Group* effects was statistically significant both at P3/P4 (F_2,55_ = 2.45, *p* = 0.044, ƞ_p_^2^ = 0.085) and at P7/P8 (F_2,55_ =3.87, *p* = 0.027, ƞ_p_^2^ = 0.123). Gamma waveforms in 2 groups (CNT and ASD) are depicted in Figure 1.

Positive area of evoked gamma oscillation to targets (TRG) was higher in ASD as compared to CNT at P7 (F_1,36_ = 7.77, *p* < 0.008); to non-target Kanizsa (NTG) at F7, F8, P3, and P4 (all *p* < 0.05); and to standard non-Kanizsa (NOK) stimuli at F8 and P7 (all *p* < 0.05). Descriptors of differences and statistical metrics are shown in Table 1. 

The TMS treatment (Figure 2) significantly decreased positive areas of evoked gamma responses to TRG at F7, F8, P7, P8; NTG at P3, P4, and P8; and NOK F7, F8, P7, P8 (all *p* < 0.05). 

Negative slope area (descending half-envelope area): *Stimulus* type had main effect at the frontal F3/F4 (F_2,55_ = 3.88, *p* = 0.023). *Stimulus* × *Group* interaction at the same site was also significant (F_2,55_ = 4.22, *p* = 0.019, ƞ_p_^2^ = 0.092). *Stimulus* had main effect at F7/F8 sites (F_2,55_ = 7.82, *p* = 0.001, ƞ_p_^2^ = 0.180) and *Stimulus* × *Group* interaction was also statistically significant (F_2,55_ = 6.36, *p* = 0.012, ƞ_p_^2^ = 0.115). Similar interaction effects were observed at the inferior parietal sites P7/P8 (F_2,55_ = 2.12, *p* = 0.039, ƞ_p_^2^ = 0.087). Figure 3 and Figure 4 illustrate these effects.

Post-hoc analysis showed following group differences: Negative area of evoked gamma oscillations at the F7 site was higher in ASD as compared to CNT to TRG (F_1,36_ = 15.85, *p* < 0.001); to NTG (F_1,36_ = 9.23, *p* < 0.001); and to NOK (F_1,36_ = 12.02, *p* = 0.001). TMS treatment decreased negative area to targets at F8, to NTG at F7, F8, P7, P8, (all *p* < 0.005) and to NOK stimuli at F7, F8, P7, P8 (all *p* < 0.05). Descriptive statistics are presented in Table 2.

Induced gamma: Positive area: No *Stimulus* × *Group* interactions were found for any frontal or parietal topographies. Negative area of induced gamma oscillations, on the other hand, showed statistical effects and group differences: *Stimulus* type had main effect both at P3/P4 (F_2,55_ = 13.8, *p* = 0.016, ƞ_p_^2^ = 0.071) and at P7/P8 sites (F_2,55_ = 9.06, *p* < 0.001, ƞ_p_^2^ = 0.251). *Stimulus* × *Group* interaction was significant at P3/P4 (F_2,55_ = 2.54, *p* = 0.044, ƞ_p_^2^ = 0.085) and at P7/P8 sites (F_2,55_ = 3.87, *p* = 0.027, ƞ_p_^2^ = 0.123). Some of these interactions are depicted in Figure 5 and Figure 6.

Negative area of induced gamma to targets was higher in ASD as compared to CNT only at P7 site (F_1,36_ = 4.97, *p* = 0.032). TMS decrease negative area of the induced gamma to NTG at P3, P4, P7, and P8 (all *p* < 0.05), and to NOK stimuli at P3 and P7 (both *p* < 0.05).

## 4. Discussion

The results of our envelope analysis show that autistic subjects as compared to neurotypicals exhibit significant differences at baseline (pre-TMS) in both evoked and induced gamma oscillations. This difference was most salient in response to non-target cues as recorded from frontal and parietal recording sites (F3, F4, F7, F8, P3, P7, P8). Post-TMS treatment, our autistic subjects exhibited a significant reduction of gamma responses to task-irrelevant stimuli. Significant changes were also observed in both the positive and negative slope areas of evoked gamma responses. The rise of the modulated waveform being higher for our ASD subjects as compared to controls and diminishing towards control levels after TMS treatment. Normalization of gamma oscillations occurred concomitantly to changes in behaviors; more specifically questionnaires (ABC, RBS-R) showed a decrease in irritability, hyperactivity, and repetitive scores. In addition, post-TMS. participants committed fewer errors (total percentage of errors and commission error percentage). 

According to pioneers of 40 Hz neurofeedback training [16] gamma response as such represents attention-related processes as it is basically a stimulus evaluation and response selection mechanism activity reflection at different stages of information processing. The early (evoked) gamma oscillation could be considered as an attention–trigger process that gives information about the arrival of a stimulus and need for more detailed processing that is occurring later in time and is reflected in cognitive event-related potentials (e.g., N200 and P300) and late (induced) gamma responses. Decrease magnitude of the evoked gamma responses to non-target stimuli post-TMS facilitates processing of targets at the later stages of the responses and is reflected in a decreased induced gamma magnitude.

Demodulation of EEG frequency bands has been used in the analysis of brainwave activity during altered states of consciousness [51]. Mathematically, it has been proven that the integration of the amplitude of the modulated EEG waveform accurately describes EEG cortical activation patterns of event-related desynchronization (ERD) [52]. Amplitude modulation usually decreases or disappears in the period prior to and at the very beginning of task performance [53]. In ASD patients, with a high autism spectrum quotient [54], ERD (ergo the integration of the amplitude modulated EEG waveform) serves to discriminate clinically significant differences such as the processing of human facial expression [55]. 

EEG sub-band modulation has been proposed as an automated non-invasive diagnostic tool for neuropsychiatric disorders such as Alzheimer’s disease wherein the slowly-varying EEG amplitudes of modulated waveforms appears to be most affected [56]. This feature, which measures the rate at which EEG sub-bands are modulated, has been called the “spectro-temporal modulation energy” [56]. TMS has been shown to change EEG activity while complex demodulation has been used to extract the power of its different bandwidths [57]. As early as the first postnatal year, abnormalities in the power of brain oscillations (delta and gamma sub-bands) help distinguish infants with ASD diagnoses from others [58]. 

In the present study, demodulation and integration of the area under the upper envelope of the gamma waveform, showed a similar effect for both the ascending and descending train of oscillations. The spectro-temporal modulation energy for both evoked and induced gamma was larger in autism than in neurotypicals and diminished towards neurotypical levels after TMS treatment. In modeling membrane potentials, a depolarizing pulse causes the interaction of parasitic properties of inductance and capacitance from some of its anatomical components. The output of the circuit is a smooth periodic sinusoidal oscillation that resonates at a particular frequency. In the absence of a driving force, the involved circuit has a tendency to vibrate around its natural frequency. It is speculated that the ability of neurons to resonate allows them to preferentially respond to those inputs arriving at the same frequency. The end result of this effect is the promotion of synchronized activity within and between neuronal populations [59]. It is believed that evoked gamma activity represents the binding of information within a confined cortical field while the induced component represents the binding across networks of different cortical regions [12,13,28]. Our results showed significant differences in both the evoked and induced components of gamma oscillations in our ASD patients as compared to neurotypicals. The findings may help explain the atypical perceptual processing symptoms observed in ASD as information processing is impeded both within and between brain regions [21]. 

In our initial study, using envelope analysis of gamma oscillations in ASD [30], a short period of ringing decay at baseline (pre-TMS) was thought to indicate a system of low sensitivity [31]; that is, one where the resonator’s center frequency in relation to its bandwidth is broadened. TMS increases the sensitivity of the system by transforming the envelope of gamma oscillations from one akin to a ramp or sawtooth waveform (baseline) to a symmetrical configuration (post-treatment). In ASD, prior to active treatment, the envelope of gamma oscillations ramps upward to a peak amplitude which is higher than in controls. This waveform is characterized by a longer latency to peak amplitude and a shorter settling time [30]. The non-sinusoidal nature of the envelope thus engendered makes it difficult to identify a dominant frequency while its sharper inflection procreates harmonic distortion. The harmonically related oscillations create additional signals that are injected back to the system and makes high-frequency EEG recordings in this patient population unreliable. TMS treatment helps transform the envelope to one, more in keeping with a sinusoidal waveform, one where spectral analysis would more easily identify a fundamental frequency that facilitates neural binding. We believe that, in ASD, the increased spectro-temporal modulation energy (vide supra) at baseline may be the result of an increased contribution of excitation to an oscillatory system that is trying to define its fundamental frequency. 

It may be reasonable to assume that abnormalities of gamma oscillation may be manifested more fully in those regions of the brain, like the prefrontal lobes, that exhibit rich and varied connectivity. Gamma oscillations play a crucial role in the binding of information between neural networks. The exceptional richness in connectivity of the frontal lobes is an important anatomical feature that may support our unique cognitive abilities as humans. Researchers describe it as the “traffic hub” of the nervous system [60]. The dorsolateral prefrontal cortex (DLPFC), a region of the prefrontal lobes, projects to a large number of cortical areas involved in visuospatial processing, auditory information and sensory integration, motor response, reward and punishment, memory, and error detection [61]. This region of the brain is most typically associated with executive functions including working memory, selective attention, mental flexibility, response initiation, impulse control, and action monitoring [62,63]. The connectivity that characterizes this area of the frontal lobes, therefore provides a low functional breakdown threshold expressed as gamma oscillation abnormalities and executive dysfunction [64]. 

Previous studies have shown that TMS targeting the frontal lobes improves executive functions [65]. These improvements have been noted on tests of cognitive flexibility, conceptual tracking, attention and working memory [65,66,67]. In ASD, TMS can change some of the core deficits, more specifically, those impairments in self-monitoring that constitute our supervisory attentional system [10,29,40,41,68]. Higher amplitude of early evoked gamma activity has been correlated to faster behavioral responses, which could constitute impulsive behaviors [69]. Indeed, results from previous trials suggest that individuals with ASD have a reduced sensitivity for monitoring errors (i.e., a diminished response time after committing an error) and instituting corrective actions. Researchers believe that a deficit in monitoring errors might manifest itself as perseverative behaviors typical of ASD [10,29,40,41,68].

In keeping with these findings TMS targeting the DLPFC in the present study provided for improvement in behaviors as noted in caregivers’ reports. The most notable change was a decrease in the T-score of the Repetitive Behavior Scale-Revised [50], along with decreased irritability and hyperactivity rating scores in the Aberrant Behavior Checklist questionnaire [49]. 

In analogy to electronic communication systems, demodulating brain oscillations allows us to examine both the envelope of a signal as well as the periodic waveform that carries the same. Gutfreund et al. [59] suggested that resonance behavior, exemplified in the carrier wave, helps tie together those neural populations that find themselves working at the same frequency. By way of contrast, the envelope of gamma oscillations could provide information regarding the impedance of involved circuits and the excitatory–inhibitory balance of the cerebral cortex [30]. Indeed, the amplitude of oscillations in the gamma frequency range, that helps define its envelope, have been shown to be dependent on the GABAergic tone [70,71,72,73]. We suggest that these metrics of gamma oscillations, those ingrained in both its carrier and its envelope, provide complementary information that allows for the better interpretation of EEG waveforms. In future studies, we hope to use these new metrics of gamma oscillations in clinical designs using different parameters of stimulation. Changes in the pulse width of the magnetic field, that better influence the recruitment of excitatory and inhibitory cells within the cerebral cortex [74], could better normalize gamma oscillation abnormalities while improving clinical outcomes of our ASD patients.

## 5. Conclusions

In this exploratory study we employed envelope analysis of demodulated waveforms for evoked and induced gamma oscillations in response to Kanizsa figures in a three-stimuli visual oddball task and compared outcomes in children and adolescents with ASD and in neurotypical peers. In ASD group, as compared to neurotypicals, significant differences in evoked and induced gamma oscillations were evident in larger gamma oscillations reflected in a higher area of gamma oscillation envelopes. Low frequency rTMS treatment in ASD resulted in a significant reduction of gamma responses to non-target stimuli and in improved accuracy of performance in the oddball task. Several rating scores of behavioral questionnaires also showed improvement post-TMS. Application of a novel metric for gamma oscillations based on envelope analysis using wavelet transformation allowed to suggest that gamma oscillations may provide a viable biomarker reflective of the excitatory/inhibitory balance of the cortex, a useful diagnostic index and a putative outcome measure of rTMS intervention in autism.

## Figures and Tables

**Figure 1 brainsci-10-00423-f001:**
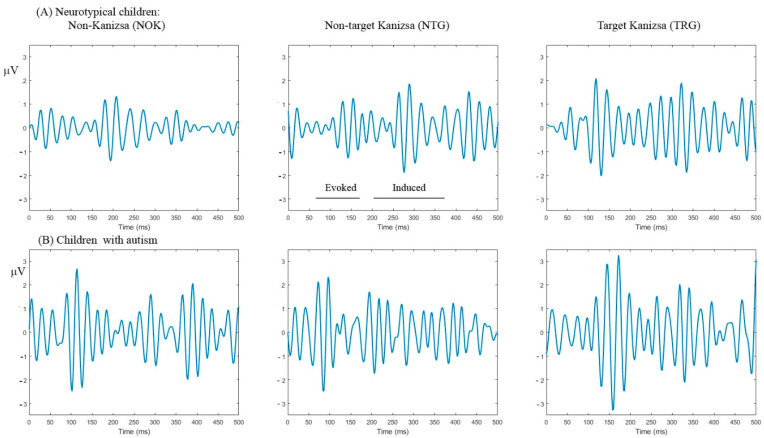
Evoked and induced gamma oscillation waveforms in response to target Kanizsa, non-target Kanizsa, and non-Kanizsa stimuli in neurotypical children (**A**) and children with autism spectrum disorder (ASD) (**B**). Both evoked and induced gamma oscillations have higher magnitude in the ASD group at the frontal sites (F3, F4).

**Figure 2 brainsci-10-00423-f002:**
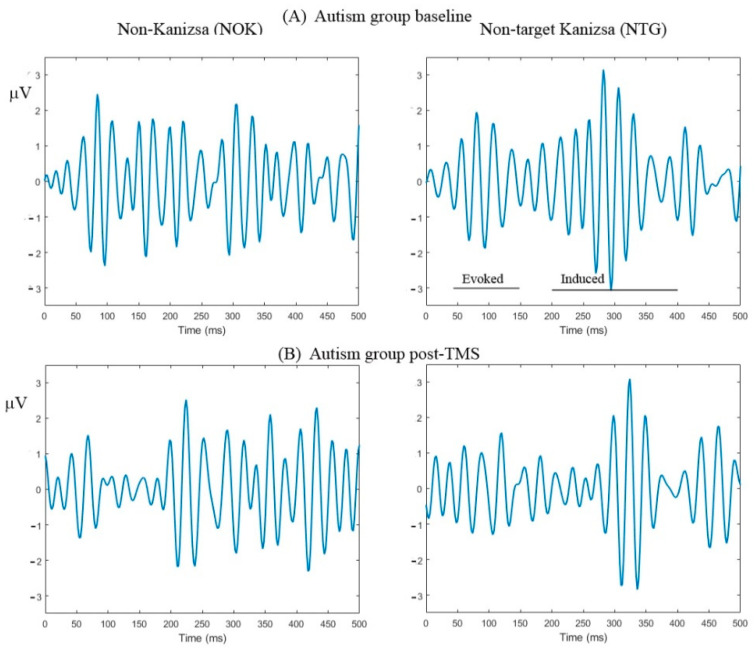
Evoked and induced gamma oscillations in response to task-irrelevant non-target Kanizsa and non-Kanizsa stimuli in children with ASD at the baseline (**A**, upper raw) and post-TMS treatment (**B**, lower raw) at the parietal sites (P7, P8). Magnitude of both evoked and induced gamma oscillations decreased post-transcranial magnetic stimulation (TMS).

**Figure 3 brainsci-10-00423-f003:**
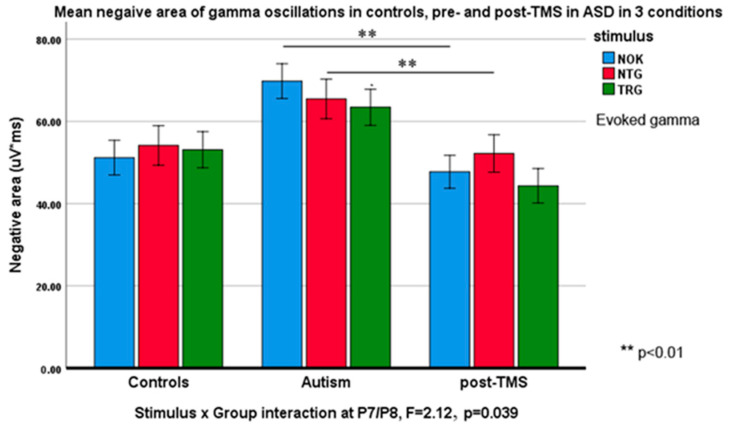
Negative slope areas (Mean ± SE) of evoked gamma oscillations in response to target Kanizsa (TRG), non-target Kanizsa (NTG), and non-Kanizsa (NOK) stimuli in typical children, children with ASD pre- and post-TMS treatment. *Stimulus* (TRG, NTG, NOK) × *Group* (CNT, ASD pre-, ASD post-TMS) interaction was significant at the frontal sites (F3, F4). Note higher area of responses at baseline and a decrease of areas in response to non-target items post-TMS.

**Figure 4 brainsci-10-00423-f004:**
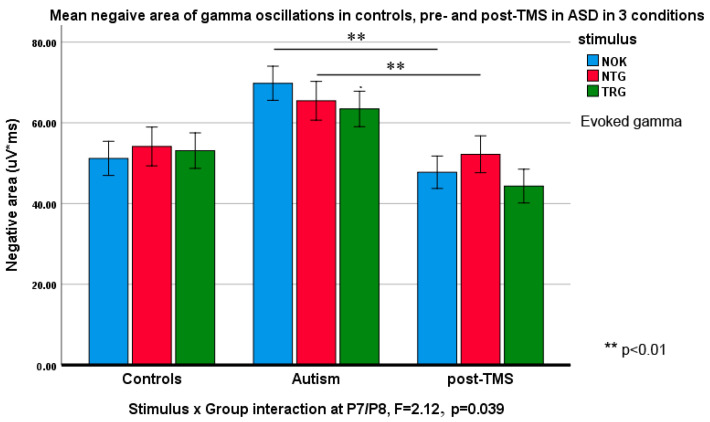
Negative slope area of evoked gamma oscillations in response to target Kanizsa (TRG), non-target Kanizsa (NTG), and non-Kanizsa (NOK) stimuli in typical children, children with ASD pre- and post-TMS treatment. *Stimulus* (TRG, NTG, NOK) × *Group* (CNT, ASD pre-, ASD post-TMS) interaction was significant at the parietal sites (P7, P8). Note higher area of responses at baseline and a decrease of areas in response to non-target items post-TMS similar to effects at the frontal sites.

**Figure 5 brainsci-10-00423-f005:**
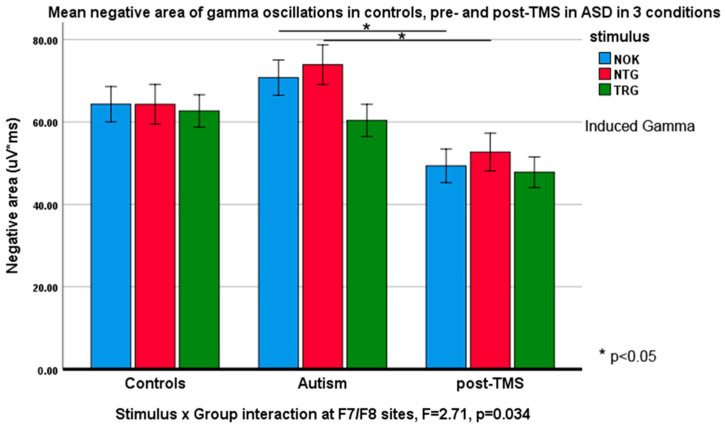
Negative slope area (Mean ± SE) of induced gamma oscillations in response to target Kanizsa (TRG), non-target Kanizsa (NTG) and non-Kanizsa (NOK) stimuli in typical children, children with ASD pre- and post-TMS treatment. *Stimulus* (TRG, NTG, NOK) × *Group* (CNT, ASD pre-, ASD post-TMS) interaction was significant at the inferior frontal sites (F7, F8). Note decrease of areas in response to non-target items post-TMS.

**Figure 6 brainsci-10-00423-f006:**
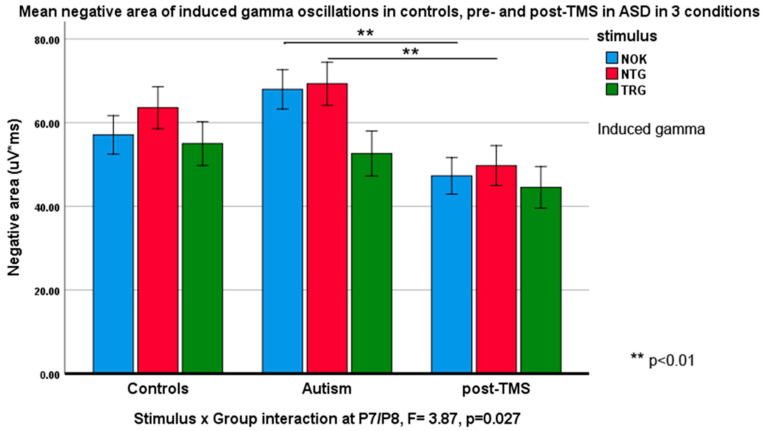
Negative slope area (Mean ± SE) of induced gamma oscillations in response to target Kanizsa (TRG), non-target Kanizsa (NTG) and non-Kanizsa (NOK) stimuli in typical children, children with ASD pre- and post-TMS treatment. *Stimulus* (TRG, NTG, NOK) × *Group* (CNT, ASD pre-, ASD post-TMS) interaction was significant at the inferior parietal sites (P7, P8). Note higher area of response at baseline and a decrease of areas in response to non-target items post-TMS similar to effects at the frontal sites.

**Table 1 brainsci-10-00423-t001:** Evoked and induced gamma oscillations positive and negative areas in response to rare target Kanizsa (Target), rare non-target Kanizsa (Non-target) and frequent non-Kanizsa standard (Non-Kanizsa) stimuli at the frontal and parietal electroencephalogram (EEG) sites in 19 children with autism (ASD) and in neurotypicals (CNT). Means ± SD. * *p* < 0.05; ** *p* < 0.001; *** *p* < 0.001.

Stimulus/EEG Site	CNT	ASD	F(1,36)	*p*
Positive/ascending area, evoked				
Target at F7	50.08 ± 22.61	69.22 ± 19.62	7.77	0.008 **
Non-target at F7	44.47 ± 13.13	59.84 ± 20.54	7.55	0.009 **
Non-target at F8	48.10 ± 15.77	60.57 ± 16.37	5.73	0.022 *
Non-target at P3	53.27 ± 19.20	73.72 ± 23.48	8.64	0.006 **
Non-Kanizsa at F7	51.56 ± 21.35	69.82 ± 27.74	5.17	0.029 *
Non-Kanizsa at P7	61.09 ± 19.23	75.92 ± 21.13	5.12	0.030*
Negative/descending area, evoked				
Target at F7	55.23 ± 4.43	49.51 ± 4.42	15.86	<0.001 ***
Non-target at F7	53.86 ± 4.55	49.21 ± 4.85	9.29	0.004 **
Non-Kanizsa at F7	61.63 ± 22.54	47.01 ± 3.12	7.84	0.008 **
Non-Kanizsa at P7	53.06 ± 21.58	68.35 ± 24.21	4.22	0.047 *
Negative/descending area, induced				
Target at P7	53.71 ± 22.95	70.30 ± 22.90	4.97	0.032 *
Non-target at P7	52.96 ± 27.07	73.67 ± 26.53	5.67	0.023 *

**Table 2 brainsci-10-00423-t002:** Effect of rTMS course on evoked and induced gamma oscillations’ positive and negative areas in response to rare target Kanizsa (Target), rare non-target Kanizsa (Non-target) and frequent non-Kanizsa standard (Non-Kanizsa) stimuli at the frontal and parietal EEG sites in 19 participants with ASD (Means ± SD). * *p* < 0.05; ** *p* < 0.001; *** *p*< 0.001.

Stimulus/EEG Site	ASD Pre-TMS	ASD Post-TMS	F(1,36)	*p*
Positive/ascending area, evoked				
Target at F7	67.25 ± 20.79	51.67 ± 18.62	6.25	0.017 *
Target at F8	74.46 ± 25.45	51.26 ± 20.97	9.97	0.003 **
Target at P7	69.22 ± 19.62	47.59 ± 19.68	12.08	0.001 **
Target at P8	56.37 ± 15.22	44.11 ± 16.69	5.82	0.021 **
Non-target at P3	73.72 ± 23.48	53.97 ± 19.95	8.27	0.007 **
Non-target at P4	76.16 ± 21.59	56.52 ± 22.92	7.74	0.008 **
Non-target at P8	66.49 ± 22.90	45.90 ± 17.67	10.25	0.003 **
Non-Kanizsa at F7	75.81 ± 25.23	48.48 ± 16.71	16.62	<0.001 ***
Non-Kanizsa at F8	75.92 ± 21.13	50.91 ± 21.58	13.66	0.001**
Non-Kanizsa at P7	69.82 ± 27.74	47.01 ± 18.62	9.49	0.004 **
Non-Kanizsa at P8	59.99 ± 25.29	44.28 ± 19.43	4.91	0.033 *
Negative/descending area, evoked				
Target at F8	48.76 ± 4.57	41.01 ± 15.70	4.29	0.045 *
Non-target at F7	51.29 ± 3.70	39.68 ± 15.57	10.03	0.003 **
Non-target at F8	54.69 ± 4.79	39.87 ± 15.38	16.17	<0.001 ***
Non-target at P7	52.46 ± 6.38	41.45 ± 15.13	8.65	0.006 **
Non-target at P8	52.07 ± 5.63	43.60 ± 16.97	4.29	0.045 *
Non-Kanizsa at F7	51.34 ± 5.25	38.80 ± 14.94	12.02	0.001 **
Non-Kanizsa at F8	51.94 ± 4.85	39.48 ± 15.63	11.07	0.002 **
Non-Kanizsa at P7	53.71 ± 5.19	40.77 ± 16.00	12.02	0.001 **
Non-Kanizsa at P8	51.13 ± 5.32	40.48 ± 15.68	7.92	0.008 **
Negative/descending area, induced				
Non-target at P3	56.80 ± 20.26	42.71 ± 16.14	5.98	0.019 **
Non-target at P4	56.37 ± 19.48	43.05 ± 18.45	4.93	0.032 *
Non-target at P7	70.30 ± 22.90	49.33 ± 19.63	9.73	0.003 **
Non-target at P8	65.66 ± 24.02	46.18 ± 22.20	7.11	0.011*
Non-Kanizsa at P3	61.07 ± 20.00	44.24 ± 20.33	6.94	0.012 *
Non-Kanizsa at P7	70.99 ± 22.65	46.82 ± 20.50	12.23	0.001 **

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
