# Peer review of "Effects of Transcranial Magnetic Stimulation Therapy on Evoked and Induced Gamma Oscillations in Children with Autism Spectrum Disorder"

_brainsci, 2020, doi:10.3390/brainsci10070423_

Round 1

Reviewer 1 Report

In their manuscript, Effects of Transcranial Magnetic Stimulation Therapy on Evoked and Induced Gamma Oscillations in Children with Autism Spectrum Disorder, Casanova et al. evaluated evoked and induced EEG gamma oscillations in both children with Autism spectrum disorder (ASD) and healthy controls. In addition to this, they evaluated how low frequency rTMS affects gamma oscillations in ASD. Their main finding is that children with ASD have larger areas in gamma oscillations and rTMS can reduce this gamma. The manuscript presents some interesting results that are of interest to the field, but major clarifications are needed.

  • What was the rationale behind the rTMS protocol? I was honestly quite surprised that such a small number of pulses could have any effect, especially, an effect lasting beyond the stimulation session.
  • What was the handedness of the participants?
  • The authors refer to the participants as children, but it seems that the participants are approximately 15 years old. Maybe “adolescents” is a better description.
  • In the abstract, the authors state that the controls were age and gender matched but, in the methods section, only age-matching is mentioned.
  • The results section in the abstract needs to be clarified. I did not understand what the authors meant by “in higher areas of gamma oscillations” before reading the rest of the manuscript.
  • Until the results section of the manuscript, I thought that also the controls received rTMS as it was never clearly stated who got what.
  • Line 188: 4-bitten should probably be 4-button
  • The authors state that “Evoked gamma was analyzed with 40-160 ms window and induced gamma within 180-400 ms window post-stimulus.” What is the rationale behind this decision?
  • The authors state that the participants underwent six weekly rTMS sessions, do they mean that treatments were given Monday-Saturday?
  • Line 266: Could the authors clarify what is meant by “The procedure for stimulation placed the TMS coil 5 cm anterior”. Are the authors referring to the motor threshold site or the treatment site?
  • Have the authors corrected the p-values for multiple comparisons?
  • What is the rationale behind the EEG sites of interest? For example, for frontal areas F3, F4, F7 and F8 were analyzed, but these are separate electrodes which do not form a continuous area around the stimulation site.
  • When the authors present results as F7/F8, do they mean that the data is sum of data from these electrodes or are the authors comparing hemispheric differences?
  • For clarity, it would be helpful if the results would be re-arranged so that first comparison between ASD and healthy controls is presented and then effects of rTMS. Also, it would be better to clarify the terms. For example, on line 308 the authors refer to post-rTMS group differences. This suggests that there are two groups that are compared after rTMS.
  • Page 7: The authors report that there are no Stimulus x Group effects or Stimulus effects at F3/F4 (which were stimulated) but all the effects of evoked gamma are on posterior sites far from stimulation. Could the authors elaborate what could be the rationale behind this?
  • Figure 1: Could the authors clarify what is presented in the figure. Average over all participants and mean of channels F3 and F4? Could data also be presented from P3/P4 and P7/P8 as these were statistically significant. The figure should also show on the time axis when the graph presents “evoked” and when “induced” activity.
  • The methods do not describe how exactly the slope area was calculated. Does it include evoked or induced activity, what is the time window?
  • Figures 3-6: Please include statistics. Also, are the error bars SD or SEM?
  • Table 1: A lot of lines are missing +/-sign
  • The authors state that artifactual trials were rejected. How many trials were included in the final analyses?
  • What was the time difference between the experimental task+EEG and rTMS? Maybe a figure showing the whole study protocol with a timeline would be helpful.

Author Response

Reviewer 1

In their manuscript, Effects of Transcranial Magnetic Stimulation Therapy on Evoked and Induced Gamma Oscillations in Children with Autism Spectrum Disorder, Casanova et al. evaluated evoked and induced EEG gamma oscillations in both children with Autism spectrum disorder (ASD) and healthy controls. In addition to this, they evaluated how low frequency rTMS affects gamma oscillations in ASD. Their main finding is that children with ASD have larger areas in gamma oscillations and rTMS can reduce this gamma. The manuscript presents some interesting results that are of interest to the field, but major clarifications are needed.

Thank you for the generally positive evaluation of our results and for suggestions to have more clarifications in the revised version of the manuscript. We believe that suggestions were very useful to improve our manuscript.

 What was the rationale behind the rTMS protocol? I was honestly quite surprised that such a small number of pulses could have any effect, especially, an effect lasting beyond the stimulation session.

The more detailed rationale for the selection of the parameters of our rTMS protocol (i.e., bilateral DLPFC as a site, 1 Hz, 180 pulses per session, 90% of MT, one session per week, number of sessions, etc.) was described in our previous journal publications [10,27-29 40,41] and in chapters and reviews (e.g., Casanova, M.F., Sokhadze, E.M., et al. Autism, transcranial magnetic stimulation and gamma frequencies. In: E.M. Sokhadze and M.F. Casanova (Eds.); Autism Spectrum Disorder: Neuromodulation, Neurofeedback and Sensory Integration Approaches to Research and Treatment. FNNR: Murfreesboro, TN, 2018, pp. 49-65.;  Casanova, M.F., and Sokhadze, E.M.: Transcranial magnetic stimulation: application in autism treatment. In: V.W. Hu (Ed.); Frontiers in Autism Research: New Horizons for Diagnosis and Treatment. World Scientific Publishing Co: Hackensack, NJ,  2014, pp. 583-606; Casanova, M.F., Sokhadze, E., Opris, I. et al. : Autism spectrum disorders: linking neuropathological findings to treatment with transcranial magnetic stimulation. Acta Pediatr. 104(4):346-355, 2015). Our protocol was guided by theoretical considerations, in particular a hypothesis that low frequency (~ 1 Hz), low power (90% of MT), and low intensity (180 pulses/per session) rTMS will activate inhibitory interneurons (e.g., double bucket cells) without activating pyramidal neurons, and that effects will result in increase of inhibitory tone in minicolumns. We proposed in our prior studies that over a course of treatment, low frequency and low power (90%  MT, low number of pulses) rTMS may restore the balance between cortical excitation and cortical inhibition by selectively activating double-bouquet cells at the periphery of cortical minicolumns. In addition, since our group was first to use rTMS in children with ASD it was decided to start with more safe mode of stimulation. Our initial pilot studies allowed to find that there were behavioral and EEG/ERP effects lasting for a week, and we continued to use this particular TMS protocol approved by the IRB and specified in our NIH-funded clinical research study protocol. In one of the latest studies we compared effects of 6, 12, and 18 sessions to demonstrate preference of using 18 session-long course of rTMS with above listed parameters of stimulation (Sokhadze at al., Front. Syst. Neurosci. 12:20, 2018; ). Effects of our protocol of rTMS on EEG was replicated recently in younger, low-functioning children with autism by Kang et al. Front Neurosci. 12:201, 2018.

    What was the handedness of the participants?

Handedness of all participants was assessed using Edinburgh questionnaire, but since no hemispheric differences in gamma oscillations were found in this study, we did not report these data in this paper. Factor Hemisphere was investigated but did not yielded any statistically significant effects.

 The authors refer to the participants as children, but it seems that the participants are approximately 15 years old. Maybe “adolescents” is a better description.

 The range of age of our participants was in 9 to 17 years (majority in 11-14 years range), this falls under NIH classification of “children”, though we agree that the mean age was better described as adolescents. In the revised version we use along with “children” also “children and adolescents” wording but specify that will continue to refer to our cohort as “children” as formally it is a correct terminology.  

In the abstract, the authors state that the controls were age and gender matched but, in the methods section, only age-matching is mentioned.

 We agree that we missed reporting the number of females in the control group (N=6), and in the revised version we confirm that the ASD and CNT group were matched by gender.

 The results section in the abstract needs to be clarified. I did not understand what the authors meant by “in higher areas of gamma oscillations” before reading the rest of the manuscript.

 Thank you for pointing on this, we revised abstract text accordingly to clarify meaning of the “higher areas of gamma oscillations”.

 Until the results section of the manuscript, I thought that also the controls received rTMS as it was never clearly stated who got what.

 This is a very important critical comment. We clarified this in our revision, and for example in describing interactions instead of “Group” we use now “Time (pre- post TMS) describing effects of rTMS in the ASD group.  We apply “Group” factor only in regards of ASD-baseline vs. CNT vs. ASD post-TMS comparisons.

Line 188: 4-bitten should probably be 4-button

           Thank you for noting this typo, we fixed it.

The authors state that “Evoked gamma was analyzed with 40-160 ms window and induced gamma within 180-400 ms window post-stimulus.” What is the rationale behind this decision?

We were guided by window range for evoked and induced gamma by classic studies such as [ 14, 18-20, 22] and had the usually used window ranges described in a review [12]. However, we extended induced gamma window up to 400 because it is well-known that induced gamma is featured by a jitter and we described in our other study the reasons why it is reasonable to have wider range for detecting late gamma oscillation peaks (see Gross et al. J. Neurother. 16(2):78-91, 2012).

 The authors state that the participants underwent six weekly rTMS sessions, do they mean that treatments were given Monday-Saturday?

 No, the sessions were conducted once per week. We described the timing of rTMS in more detail in our revision.

Line 266: Could the authors clarify what is meant by “The procedure for stimulation placed the TMS coil 5 cm anterior”. Are the authors referring to the motor threshold site or the treatment site?

  We followed recommendation by [42, 43] to select the site for rTMS. It was illustrated by a figure in our Sokhadze at al.,  Front. Syst. Neurosci. 12:20, 2018 paper. The coil was placed 5 cm anterior of motor strip. Motor threshold was detected by stimulation of the motor strip and procedure is now described in the paper.

Have the authors corrected the p-values for multiple comparisons?

We reported only Greenhouse-Geisser (G-G) corrected p-values when multiple comparisons were used.  Eventually, in some cases of our statistical result reports it resulted in slightly higher p-values (e.g., 0.043 after GG correction rather than 0.038 without correction). We confirm in our Methods section that corrections were applied when appropriate to do so.

 What is the rationale behind the EEG sites of interest? For example, for frontal areas F3, F4, F7 and F8 were analyzed, but these are separate electrodes which do not form a continuous area around the stimulation site.  Also:  Page 7: The authors report that there are no Stimulus x Group effects or Stimulus effects at F3/F4 (which were stimulated) but all the effects of evoked gamma are on posterior sites far from stimulation. Could the authors elaborate what could be the rationale behind this?

Effects of rTMS are not limited to the stimulated target cortex (in our protocol it was DLPFC)  because of functional and anatomical  interconnections of cortical areas within a distributed functional network (Rossi and Rossini 2004; Ziemann 2004).  The large output territory of the DLPFC makes it a connection hub within the small-world network of corticocortical connectivity. It was thought that modulating the output of the DLPFC would procreate a beneficial cascading to the many areas connected to the same, as we noted in our prior publications (e.g., Casanova et al., Acta Pediatrica, 104(4):346-355, 2015)

    When the authors present results as F7/F8, do they mean that the data is sum of data from these electrodes or are the authors comparing hemispheric differences?

 The statistical analysis model used for evaluation of these two frontal sites (F7 and F8) was for ASD vs. CNT -  Stimulus (target Kaniszsa, non-target Kanizsa, non-Kanizsa Standard)  x Hemisphere (F7, F8) x Group (ASD-baseline, CNT, TMS) and for rTMS effects only it was Stimulus x Hemisphere (F7,F8) x Time (ASD pre-, ASD post-TMS). We were trying to find effects of hemisphere and expected lateralization effects but could not find any statistically significant lateralization of effects. Thus, reporting F7/F8 (and in a similar manner F3/F4, P3/P4, or P7/P8) we have summary of F7 and F8 outcomes. In this study we did not analyze more widely defined regions-of-interest such as F3 plus F7 vs. F4 vs. F8.

For clarity, it would be helpful if the results would be re-arranged so that first comparison between ASD and healthy controls is presented and then effects of rTMS. Also, it would be better to clarify the terms. For example, on line 308 the authors refer to post-rTMS group differences. This suggests that there are two groups that are compared after rTMS.

 We were considering initially re-arranged comparison of ASD vs CNT and  reports of pre- post-TMS effects in the ASD group, however, later decided to leave the order we selected (i.e., initial group [CNT, ASD,TMS] differences analysis followed  by post-TMS effects) to avoid substantial reshuffle of the illustration order and a need of additional descriptions of topography and type of gamma response (evoked and induced). We again outline that only ASD group was enrolled into rTMS treatment. Neurotypical children were tested only once for the purposes of comparing with ASD metrics at baseline and post-TMS.

 Figure 1: Could the authors clarify what is presented in the figure. Average over all participants and mean of channels F3 and F4? Could data also be presented from P3/P4 and P7/P8 as these were statistically significant. The figure should also show on the time axis when the graph presents “evoked” and when “induced” activity.

We followed recommendation of the reviewer by adding evoked and induced labels to figures on time axis.  We decided to present only most representative pairs of EEG channels. We discussed feasibility of added  more figures (e.g., P3/P4, P7/P8) per reviewer’s request but decided not to add more bar figures. The other reviewer complained about too many figures in the paper, so we tried to balance among these critics.

The methods do not describe how exactly the slope area was calculated. Does it include evoked or induced activity, what is the time window?

 We added description of the slope calculation details in the Methods section. The calculation of the slope area (ascending and descending halves of the envelope) is more clearly described in the revised version. Eventually it is half-envelope area calculated for instance for ascending slope as amplitude by latency divided by two. Methods of latency and amplitude calculations are described in detail and include relevant equations.

Figures 3-6: Please include statistics. Also, are the error bars SD or SEM?

  The bas on the figures are Standard Errors (SE) and this is now clearly specified in the legends to figures. In the statistics description (and in tables) we used Standard Deviations (SD) only.

    Table 1: A lot of lines are missing +/-sign

               We apologize for these typos and included all missing plus/minus signs.

The authors state that artifactual trials were rejected. How many trials were included in the final analyses?

The EEG segments were analyzed on trial by trial basis. Trials with artifacts were rejected. There were initially 30 trials for target Kanizsa, 30 for non-target Kanizsa and 180 trials for non-Kanizsa stimuli. By the previously set rule, at least 20 clean trials were considered as sufficient for target and non-target Kanizsa (i.e., at least 20 per each of those stimuli) and at least 60 successful trials for non-Kanizsa stimuli.

 What was the time difference between the experimental task+EEG and rTMS? Maybe a figure showing the whole study protocol with a timeline would be helpful.

Baseline experimental oddball task was conducted before the start of rTMS sessions, usually at the separate visit within a week before the rTMS course, while post-TMS oddball EEG task was conducted within a week after completion of the 18-session long rTMS course, but not earlier than a day after the completion of the course (and not later than a week after completion of the course). Adding additional flow-chart for this relatively simple design we consider as an excessive.

Reviewer 2 Report

REVIEW on paper:

Effects of Transcranial Magnetic Stimulation Therapy on Evoked and Induced Gamma Oscillations in Children with Autism Spectrum Disorder

Thank you for the opportunity to review this interesting paper. The authors investigated the EEG gamma oscillations as a biomarker in ASD using a novel metric, envelope analysis of demodulated waveforms, for evoked and induced gamma oscillations in response to Kanizsa figures in an oddball task. The subjects were treated with low frequency rTMS for 18 weeks, and the effect was assessed on both the gamma oscillations, error rate as well as with behavioral changes assessed using two different questionnaires (ABC and RBS-R) using caregiver reports. The results suggest gamma oscillations as a biomarker reflective of the excitatory/inhibitory balance of the cortex.

The study is well-planned and performed with sound basis. The paper is well-written and easy to follow with clear aims and this is adding to the current knowledge. The language is good standard English though there are a few typos (missing of spaces and spelling of envelope). The abbreviations can be used throughout the text after they have been opened up (DLPFC, ABC, RBS-R). The references are relevant and up-to-date. However, I have some specific comments.

Abstract: The pulse number in rTMS could be stated. What is meant with higher areas of gamma oscillations? It is not defined.

Introduction: At the end of the Introduction, please define the previous findings that this study expands.

Methods: Was the ADHD accepted as co-diagnosis? Are WISC-IV and WASI comparable to each other? Can you specify the university or the number of the IRB approval? Is the word record correct on page five, line 199, or should it be ‘accepted’? What was the time windows for evoked and induced based on, is there any reference? In the editing of this version, the special characters are missing in the chapter 2.3.1, there are empty spaces. The pulse number is very low for each of the treament session, only 180. What was this based on? Was the MT checked before each treatment? There should be a reference to placing the coil 5 cm anterior. Is this corresponding to F3 electrode site?

Results: Could you have a table on the behavioral outcome in total and for subscales? Now the results are shown for certain subscales. How were these chosen? Could you have separate images for evoked and induced responses or mark it in the figures? Significant differences should be visible in all of the figures. Is there any more informative way of expressing the results in Figures 3, 4, 5 and 6 than bar diagram? I think some important information is now missing. A table might be useful. In the manuscript there are so manyt tables and figures, that it is difficult to get the point. As well, the statistically significant findings should be marked with asterisks. There are some plus-minus missing in Table 1. Are the degress of freedom stated or not? It only shows the statistically significant results. Maybe the non-significant results could be given as supplementary material? The terms rare and frequent appeared quite suddenly in the results. This should be clearly specified in the Methods (compare 2.1) so it is easy to get here. Is there any more measure you could use? For example latency to peak amplitude. Some of the results were depicted in the discussion, but not actually shown in the results.

Discussion: How is the neurofeedback nowadays? Is there anything more recent that relates to this study you could refer to? The paragraph on page 14 (lines 471-483) might be more appropriate in the Introduction. Similarly, the paragraph on carrier wave is kind of important and could be introduced earlier in the paper (lines 498-507). TMS targeting the frontal lobes means rTMS, and this could be specified as low- or high-frequent.

Author Response

Reviewer 2

Thank you for the opportunity to review this interesting paper. The authors investigated the EEG gamma oscillations as a biomarker in ASD using a novel metric, envelope analysis of demodulated waveforms, for evoked and induced gamma oscillations in response to Kanizsa figures in an oddball task. The subjects were treated with low frequency rTMS for 18 weeks, and the effect was assessed on both the gamma oscillations, error rate as well as with behavioral changes assessed using two different questionnaires (ABC and RBS-R) using caregiver reports. The results suggest gamma oscillations as a biomarker reflective of the excitatory/inhibitory balance of the cortex.

      We appreciate detailed summary of our findings and considering our paper as an interesting one.

The study is well-planned and performed with sound basis. The paper is well-written and easy to follow with clear aims and this is adding to the current knowledge. The language is good standard English though there are a few typos (missing of spaces and spelling of envelope). The abbreviations can be used throughout the text after they have been opened up (DLPFC, ABC, RBS-R). The references are relevant and up-to-date. However, I have some specific comments.

Thank you for positive evaluation and for providing very specific comments that helped to improve quality and clarity of our manuscript.

Abstract: The pulse number in rTMS could be stated. What is meant with higher areas of gamma oscillations? It is not defined.

This concern was noted as well by the other reviewer and we addressed it by more detailed explanation of the meaning of the “high area of gamma oscillations”.

Introduction: At the end of the Introduction, please define the previous findings that this study expands.

We added several phrases in the Introduction referring to our prior studies using rTMS in ASD population and explained how the current study adds to our prior findings.

Methods: Was the ADHD accepted as co-diagnosis?

Majority of our subjects with autism were diagnosed using DSM-IV and ADHD comorbidity was not allowed in that manual. Only 4 of the subjects were initially diagnosed with ADHD but later were more correctly diagnosed by experienced pediatric psychologist as high-functioning autism and/or Asperger Syndrome. One subject was diagnosed using DSM-5 and had comorbid ADHD in his medical record. In our later studies in younger study participants diagnosed using DSM-5 we see significantly higher rate of comorbidity od ASD and ADHD. We described this phenomenon in our recent publication on ERP differences between children with ASD, ADHD, comorbid ASD+ADHD and neurotypical children (Sokhadze et al. NeuroRegulation 6(3):134-152, 2019).  This question is very relevant. We added in our cohort description that 5 autistic children had ADHD comorbidity.

Are WISC-IV and WASI comparable to each other?

We accepted both WISC-IV and WASI according to records provided by our pediatric collaborators. In some cases selection of one of above instruments was determined by the age of subjects and training of clinical psychologist conducting evaluations of IQ. It was nor feasible for our study to try to have all participant being diagnosed by the same diagnostic instrument. Our main concern was to be assured that the subject will understand the task and will comply with oddball test and TMS procedure requirements.

Can you specify the university or the number of the IRB approval?

The protocol of the study was reviewed and approved by the University of Louisville IRB (protocol number 006.07 ). We do mention now the name of institution IRB that approved the protocol but consider listing of the IRB protocol number as excessive (as it had several amendments).

 Is the word record correct on page five, line 199, or should it be ‘accepted’?

          We replaced word “recorded” with “accepted” instead, as recommended.

What was the time windows for evoked and induced based on, is there any reference?

Selection of windows was following recommendations from [ 18-20] and our review on this topic [12]. We addressed the issue of expanding window for induced gamma due to jitter effects reviewed in more detail in our publication that used alignment method for more correct identification of the peak of late gamma oscillations (Gross et al. J. Neurother. 16(2):78-91, 2012).

The pulse number is very low for each of the treament session, only 180. What was this based on?

We responded to this question that was also posed by the reviewer #1. In particular we stated that our protocol was guided by theoretical considerations, in particular a hypothesis that low frequency (~ 1 Hz), low power (90% of MT), and low intensity (180 pulses/per session) rTMS will activate inhibitory interneurons (e.g., double bucket cells) without activating pyramidal neurons, and that effects will result in increase of inhibitory tone in minicolumns. In addition, since our group was first to use rTMS in children with ASD it was decided to start with more safe mode of stimulation. Our initial pilot studies allowed to find that there were behavioral and EEG/ERP effects lasting for a week, and we continued to use this particular TMS protocol approved by the IRB and specified in our NIH-funded clinical research study protocol. In one of the latest studies we compared effects of 6, 12, and 18 sessions to demonstrate preference of using 18 session-long course of rTMS with above listed parameters of stimulation (Sokhadze at al., Front. Syst. Neurosci. 12:20, 2018). Effects of our protocol of rTMS on EEG was replicated recently in younger, low-functioning children with autism by Kang et al. Front Neurosci. 12:201, 2018.

Was the MT checked before each treatment?

The Motor Threshold was checked before the start of the course at the left hemisphere (session #1), at session #7 at the right hemisphere, and at the session #13 at both left and right hemispheres. The outcomes of MT test were used to determine power (90% of MT) for the left and right hemispheres. In several cases we had clear differences in MT for the left and right hemispheres. We added sentences to specify timing of MT detection for each hemisphere.

There should be a reference to placing the coil 5 cm anterior. Is this corresponding to F3 electrode site?

Yes, according to [42,43] the topographic locations are corresponding to F3 (for the left DLPFC) and F4 (for the right DLPFC).

Results: Could you have a table on the behavioral outcome in total and for subscales? Now the results are shown for certain subscales. How were these chosen?

The focus of our study was on EEG metrics rather than behavioral outcomes, we reported some of behavioral outcomes to demonstrate effects of rTMS not only on EEG but also behavioral measures. ABC has 5 subscales and RBS-R also several subscales. We reported only subscales rates that did show significant statistical changes. We did not report non-significant changes in our EEG  outcomes, so it was logical to report only statistically significant behavioral changes post-TMS. However, following suggestion of the reviewer we added outcomes of several sub-scales rating that showed positive trends. Eventually, from prior studies we were aware what particular subscale is expected to show positive changes and were guided by our own experience. But again, this was not a clinical trial, it was a research study seeking for usefulness of gamma EEG biomarkers to explain neuropathology o ASD (e.g., excessive E/I ratio) and potentially positive effects of rTMS on cortical overexcitation in ASD.

Could you have separate images for evoked and induced responses or mark it in the figures? Significant differences should be visible in all of the figures.

We added marked evoked and induced gamma windows on 2 of our figures. We tried to select figures showing visible differences matching our statistical results.

Is there any more informative way of expressing the results in Figures 3, 4, 5 and 6 than bar diagram? I think some important information is now missing. A table might be useful. In the manuscript there are so many tables and figures, that it is difficult to get the point.

Initially we planned to have these results interactions (e.g., those available from SPSS figures) but later changed our mind and decided to go for bar figures. We had to present gamma waveforms to show the oscillations and we had to do tables of data relevant to EEG sites that did show interaction (and eligible for post hoc analysis). In this manuscript, the changes of the type of illustrations and tables could result in too substantial remake and multiple revisions of both illustrations and text of the paper, so we were reluctant to go for such major changes that could distort integrity of the focus and the primary message of our paper. The manuscript already has too many tables and figures, adding more might over-complicate it even further.

 As well, the statistically significant findings should be marked with asterisks.

       We marked significant differences with asterisks per reviewer’s recommendation.

There are some plus-minus missing in Table 1.

       We fixed missing signs, thank you for noting these typos.

 Are the degrees of freedom stated or not? It only shows the statistically significant results.

Degrees of freedom are listed now when reporting significant results per reviewer’s suggestion. All F values are accompanied by degrees of freedom.

Maybe the non-significant results could be given as supplementary material?

Considering the number of the metric analyzed (including amplitude of the peaks of evoked and induced gamma, latency of evoked and induced gamma peaks, latency of ascending and descending  slopes, areas of ascending and descending slopes at all EEG site, differences between ascending and descending latencies and areas) it would not be feasible to create such overwhelmingly excessive supplemental material for this exploratory study. When we will go for a controlled rTMS study in ASD we will definitely consider creating supplemental materials. In this particular study we consider this as an excessive effort.

The terms rare and frequent appeared quite suddenly in the results. This should be clearly specified in the Methods (compare 2.1) so it is easy to get here.

We apologize for using jargon typical for oddball studies. In the Method section we clearly define target Kanizsa stimulus (TRG) as a rare (uncommon) stimulus, non-target Kanizsa stimulus (NTG) as rare, task-irrelevant stimulus, and non-Kanizsa as a frequent (common) standard stimulus (NOK) and now keep referring to certain stimuli in more unified terminology. Both non-target Kanizsa and non-Kanizsa stimuli are referred to as task-irrelevant stimuli.

Is there any more measure you could use?  For example latency to peak amplitude. Some of the results were depicted in the discussion, but not actually shown in the results.

We analyzed amplitude of the peaks of evoked and induced gamma, latency of evoked and induced gamma, latency of ascending and descending  slopes, areas of ascending and descending slopes at all EEG site, differences between ascending and descending latencies and areas all above for evoked and induced gamma oscillations to targets, non-target Kanizsa and for non-Kanizsa stimuli. We included in the reported outcomes only those measures that showed consistent statistical differences, in particular Stimulus x Hemisphere x Group interaction and significant post-hoc analysis results.

Discussion: How is the neurofeedback nowadays?  Is there anything more recent that relates to this study you could refer to?

The combined use of rTMS and EEG neurofeedback has been used to operantly condition post-TMS EEG changes in our prior study (Sokhadze at al. Appl. Psychophysiol. Biofeedback 39(3-4):237-257, 2014).  The underlying hypothesis was that combined TMS and neurofeedback therapy would be synergic and improve executive functions and behavior in the treatment group (n = 20) as compared to the wait list group (n = 22). Results of the integrated neuromodulation treatment supported the initial hypothesis by demonstrating significant improvements in the behavioral and ERP measures of executive functions, as well as significant changes in EEG outcomes of neurofeedback training such as frontal theta-to-beta ratio and an increased relative power of gamma activity.  We consider this combination of rTMS and neurofeedback as a very potentially powerful approach. However, we submitted several proposals on this integrated treatment but could not secure funds for continuation of that project. On the other hand, we had neurofeedback in autism (and in ASD with comorbid ADHD) study funded and running. We are presenting these results at neurofeedback society meeting (ISNR 2020) and had a chapter accepted for publication.

The paragraph on page 14 (lines 471-483) might be more appropriate in the Introduction. Similarly, the paragraph on carrier wave is kind of important and could be introduced earlier in the paper (lines 498-507).

  We introduce information about carrier wave and envelope in the Introduction as recommended. We wrote relevant paragraphs in the Introduction.

TMS targeting the frontal lobes means rTMS, and this could be specified as low- or high-frequent.

          We refer TMS as rTMS and specify it as a low-frequency inhibitory stimulation as recommended.

Round 2

Reviewer 1 Report

No further comments.